# Research on Improved GRU-Based Stock Price Prediction Method

Chi Chen [ID], Lei Xue and Wanqi Xing *

China School of Communication and Information Engineering, Shanghai University, Shanghai 200444, China;
chenc@shu.edu.cn (C.C.); 16301098@163.com (L.X.)
*   Correspondence: xingwq@shu.edu.cn

**Abstract:** The prediction of stock prices holds significant implications for researchers and investors evaluating stock value and risk. In recent years, researchers have increasingly replaced traditional machine learning methods with deep learning approaches in this domain. However, the application of deep learning in forecasting stock prices is confronted with the challenge of overfitting. To address the issue of overfitting and enhance predictive accuracy, this study proposes a stock prediction model based on a gated recurrent unit (GRU) with reconstructed datasets. This model integrates data from other stocks within the same industry, thereby enriching the extracted features and mitigating the risk of overfitting. Additionally, an auxiliary module is employed to augment the volume of data through dataset reconstruction, thereby enhancing the model's training comprehensiveness and generalization capabilities. Experimental results demonstrate a substantial improvement in prediction accuracy across various industries.

**Keywords:** stock prices prediction; gated recurrent unit; overfitting; reconstructed datasets



## 1. Introduction

With the development and gradual refinement of the corresponding systems in Chinese stock markets, an increasing number of individuals have grown interested in participating in stock market investment. However, due to the fact that stock prices are influenced by various factors, such as policy adjustments and company performance [1], which are themselves highly unstable, accurately predicting the future trends for stock prices is crucial to help investors achieve higher returns and better manage potential risks. Furthermore, predicting stock prices can also help enterprises make better investment decisions, ultimately increasing their value and profitability. Therefore, predicting the future trends for stocks has become one of the most attractive research topics in the academic community.

There are various methods for predicting stocks, which can be broadly classified into fundamental analysis and technical analysis. Currently, technical analysis methods commonly used in China and abroad can be roughly divided into two categories: econometric methods and machine learning (ML) methods. The mainstream econometric models, such as the autoregressive moving average (ARMA) [2] model, the autoregressive integrated moving average (ARIMA) [3] model, the generalized autoregressive conditional heteroscedasticity (GARCH) model, the vector auto regression (VAR) model, and so on, have been proven effective in predicting the stock market according to the literature. Although econometric methods are more objective in nature and supported by appropriate theories, their effectiveness with regard to stock market prediction relies on the strictness of their underlying assumptions, and they are only applicable to linearly structured data. However, given that the stock market is a dynamic system influenced by various factors and often characterized by a series of complex and nonlinear features, traditional econometric prediction methods are restricted by certain limitations and are not well suited to the analysis of complex, high-dimensional, and noisy financial time series.

To achieve better results, some complex and nonlinear ML methods, such as support vector machines (SVMs) [4], genetic algorithms (GAs) [5], fuzzy logic (FL) [6], and hybrid models [7], have been widely used by researchers in stock price prediction. Compared to traditional econometric methods, ML requires fewer assumptions and has a significant advantage in extracting data features, thus making it able to handle nonlinear and non-stationary data. In recent years, deep learning models have replaced enhanced machine learning methods in stock market forecasting. However, in stock prediction, due to the limited amount of historical price data available for training, some deep learning models, such as convolutional neural networks (CNNs) [8] and long short-term memory (LSTM) [9], tend to overfit. The purpose of this paper is precisely to address this problem. The main research content of this paper is as follows:

1.　A stock prediction framework is proposed, employing data augmentation methods to expand the dataset and mitigate the risk of overfitting;
2.　The performance of the model was validated by employing real stock data from several industries in China and it showed superior outcomes and reduced errors compared to existing methodologies, thus making it possible to enhance the accuracy of stock prediction.

In this study, we introduce an enhanced model based on a GRU to forecast stock price trends by incorporating key factors that influence stock prices, such as industry trends. Our objective was to enhance the performance of the model and minimize prediction errors. Furthermore, we constructed a refined dataset by integrating data from other stocks within the same industry with the dataset for the target stock to improve the accuracy of stock price prediction. A comparative analysis of our proposed approach against the performance of the GRU model demonstrated its superior predictive capabilities and reduced margin of error. The novelty of our research lies in the unique application of industry-wide stock data, which can capture comprehensive industry trends and distinctive features. The augmented dataset not only mitigates the risk of model overfitting but also significantly enhances the precision of stock price forecasting for the target stock.

## 2. Related Work

Scholars in China and abroad have engaged in extensive exploratory studies pertaining to the utilization of machine learning in stock price prediction. In the context of utilizing a backpropagation neural network (BPNN) for stock price prediction, Wu et al. [10] applied it to predict the ups and downs of the Shanghai Composite Index, and the results indicated that the model was effective in predicting the Chinese stock market. Ticknor [11] subsequently employed the method to predict the trends for Microsoft and Goldman Sachs stock prices, confirming its effectiveness. Additionally, Zhang et al. [12] employed it for stock price prediction and achieved a notable accuracy rate of 73.29% through empirical testing. Tay et al. [13] studied the application of support vector regression (SVR) in stock market prediction, demonstrating the superiority of SVR in stock market prediction. Ran et al. [14] used a BPNN and SVR to construct a stock price prediction model, and the results showed that the SVR stock price prediction model had smaller errors and higher accuracy when predicting stock price trends. Kim [15] used support vector machines (SVMs) to classify the daily directional changes in the Korean stock market index (KOSPI) and compared the results with those of neural networks (NNs) and case-based reasoning (CBR) predictions, showing that SVMs had better predictive performance. However, shallow ML algorithms possess relatively simple structures and may exhibit insufficient handling capabilities for raw data. Moreover, such algorithms are frequently susceptible to issues such as local optima or overfitting and may experience slow convergence during real-world application scenarios.

To address the above issues associated with ML, researchers have resorted to the application of deep learning methods for stock price prediction. Deep learning, proposed by Hinton et al. [16], has been widely adopted in modeling time-series data. Singh et al. [17] used a deep neural network (DNN) to predict the NASDAQ index, achieving 17.1% higher

accuracy in comparison to a radial basis function neural network (RBFNN), demonstrating that deep learning can enhance the accuracy of stock price prediction.

Kraus et al. [18] integrated a DNN, gradient-boosted trees, and random forests to predict the future returns of the S&P500 index stocks over a selected time period. To predict high-frequency stock market trends, Chong et al. [19] combined a DNN with three unsupervised feature extraction methods: principal component analysis (PCA), an autoencoder, and a restricted Boltzmann machine. Cui [20] used deep belief networks (DBNs) to prognosticate future stock price changes, recording better performance in comparison to BPNNs and RBFNNs. Similarly, Liu [21] combined fuzzy theory with a DBN to propose a fuzzy deep-learning network model for stock price prediction, which exhibited satisfactory prediction performance and broad research prospects from the experimental results. Li et al. [22] introduced intrinsic plasticity into a DBN, enabling the model to have adaptive capabilities, and the results showed that the prediction accuracy for stock closing prices was significantly improved. Tsantekidis et al. [23] encoded sequence data with an encoder and then used a CNN for prediction, demonstrating that the CNN was better suited for predicting stock trends compared to other methods, such as the MLP and SVMs. Sim et al. [24] established a CNN-based stock price prediction model for the S&P500 index and compared the accuracy of the model with artificial neural networks (ANNs) and SVR, and the experimental results showed that the CNN is an ideal choice for developing stock price prediction models. Furthermore, Chen et al. [25] proposed a CNN-based stock trend prediction model dependent on graph convolutional features and verified the superiority of the model using six randomly selected Chinese stocks. Additionally, Persio et al. [26] utilized the multilayer perceptron (MLP) and convolutional neural networks (CNNs) to predict the opening and closing prices of the S&P500 index on the next day and concluded that CNNs exhibited smaller prediction errors compared to the MLP. Hsieh et al. [27] first utilized the wavelet transform to decompose stock prices for noise elimination and then used recurrent neural networks (RNNs) optimized with an artificial bee colony algorithm to predict stock prices in real time. Rather et al. [28] proposed a hybrid predictive model comprising autoregressive moving average models, exponential smoothing models, and an RNN to predict stock returns, which showed better prediction performance than a single RNN. Qin et al. [29] proposed a double-stage attention-based RNN model that adaptively extracts relevant input features for prediction and showed that the model is more effective in stock dataset prediction than other techniques.

To mitigate the widespread challenges of gradient vanishing or exploding and long-term dependencies in neural networks, the long short-term memory (LSTM) neural network was proposed by Hochreiter et al. [30] and has been widely used for time-series prediction. Compared with traditional RNNs, LSTM is better able to solve the problem of long-term dependencies by retaining information previously processed during training. Persio et al. [26] compared the performances of an RNN, LSTM, and a GRU in the prediction of Google stock prices and found that the LSTM neural networks had advantages in stock price prediction. Yang et al. [31] extended their research to 30 global stock indices and constructed an LSTM model to compare short-term, medium-term, and long-term prediction performance. According to the results, the LSTM demonstrated higher prediction accuracy compared to the econometric method ARIMA and the SVR and MLP ML methods across all indices for different periods. Deep learning has been proven to produce highly accurate predictions across a broad range of applications. However, the serious issue of overfitting [32] is a significant concern in deep learning, particularly when the training dataset is small relative to the complexity of the model. In such situations, the deep learning model may memorize the training data rather than generalize to new inputs, leading to poor performance with unseen test data. Given the limited availability of historical data that the stock market prediction can rely on, such models are prone to a higher risk of overfitting. Despite the availability of various regularization techniques to mitigate overfitting, the problem remains a significant challenge in deep learning, and ongoing research is focused on developing more effective solutions to this critical issue.

In previous studies, data augmentation methods in the context of deep learning were primarily based on target-specific data. However, in this paper, we propose the utilization of mixed data from other stocks within the industry to fine-tune the training model, allowing the model to learn the entire industry's characteristics and thereby reducing the risk of overfitting.

## 3. Materials and Methods

Time-series index data are themselves an important and direct source of bias in predicting stock market indices. A simple model involves utilizing historical target data as input to forecast future movements. The left side represents the input comprising historical stock data, and the output is future stock prices. However, deep learning models require a large amount of data to make effective predictions. In the example presented in this paper, the number of data points was less than 10,000, which can lead to the problem of overfitting, where the model is excessively trained to achieve high training accuracy but low testing accuracy. To mitigate the problem of overfitting, it is imperative to augment the relevant dataset without altering the original data. This augmentation can be based on the concept of incorporating an auxiliary module, as proposed in Section 3.2. This auxiliary module employs a restructured relevant dataset to assist the prediction module in making accurate predictions.

### 3.1. GRU

This paper focuses on time series prediction in particular, for which the commonly used deep learning method is the RNN method. However, RNNs can encounter issues such as gradient explosion and vanishing, particularly when learning long-term dependencies in the data. To resolve these problems, studies have proposed LSTM, which improves the gradient flow within a network by employing a gating mechanism. The GRU is a simplified version of LSTM, reducing the three gates in LSTM to two. Consequently, the GRU exhibits enhanced proficiency in capturing and learning long-term dependencies in time-series data while also reducing model complexity and computational costs, thus providing superior training efficiency. The improved ability of the GRU to handle long-term dependencies in time-series data makes it the preferred choice. Additionally, the GRU has lower storage requirements, rendering it suitable for processing large-scale datasets. Therefore, the basic GRU model was selected as the primary model in this study. The architecture of the GRU model is illustrated in Figure 1.

$$r_t = \sigma(W_r x_t + U_r h_{t-1} + b_r) \tag{1}$$

$$z_t = \sigma(W_z x_t + U_z h_{t-1} + b_z) \tag{2}$$

$$\widetilde{h_t} = \tanh(W_h x_t + U_h(r_t * h_{t-1}) + b_h) \tag{3}$$

$$h_t = z_t * h_{t-1} + (1 - z_t) * \widetilde{h_t} \tag{4}$$

Here, * represents the element-wise product formula; $W_r$ and $W_z$ are the weight matrices of the $r_t$ gate and the $z_t$ gate, respectively; $U_h$ represents the weight matrix for the output; $x_t$ represents the input data at time $t$; $\widetilde{h_t}$ and $h_t$ represent the candidate state and output state at time $t$; $b_r$, $b_z$, and $b_h$ are constants; and σ and tanh are the sigmoid and tanh activation functions, respectively, used to activate the control gates and candidate states.

After the information enters the GRU unit, the process of flow transmission includes the following steps:

(1) The input data $x_t$ at time $t$ and the output of the hidden layer $h_{t-1}$ at time $t-1$ are concatenated. The output signal of the reset gate $r_t$ is obtained with Formula (1);

(2) The output signal of the update gate $z_t$ is obtained with Formula (2);

(3)  The current-state hidden-unit candidate set $\tilde{h}_t$ is obtained with Formula (3), which mainly integrates the input data $x_t$ and the hidden layer state at time $t-1$ after filtering by the reset gate;

(4)  The output of the hidden layer $h_t$ at time $t$ is obtained with Formula (4), which represents the forgetting of the hidden layer information $h_{t-1}$ passed at time $t-1$ and the selection of important information from the candidate hidden layer at time $t$.

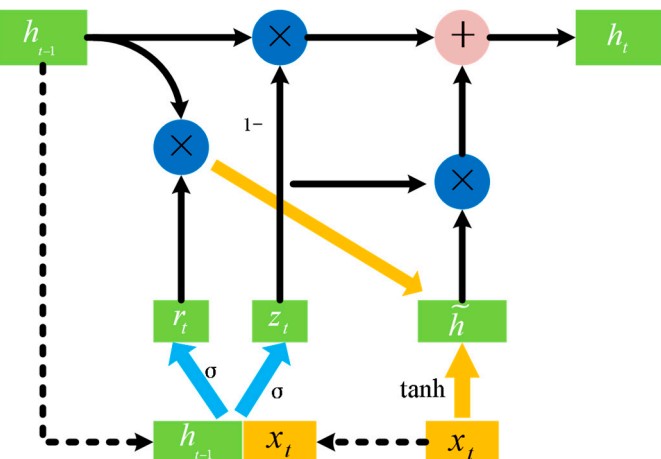

**Figure 1.** GRU architecture block diagram.

### 3.2. The Proposed Model Architecture

To address the problem discussed above, this paper proposes a combined model, as illustrated in Figure 2, based on the GRU algorithm. The left module takes the historical data for the target stock to be predicted as input and uses a GRU module to process it. The right module serves as the auxiliary module, which takes input data constructed using the approach described in Section 3.5. Its function is to fine-tune the left prediction module by incorporating features related to an industry relevant to the target stock, thereby avoiding overfitting and improving the effectiveness of the prediction.

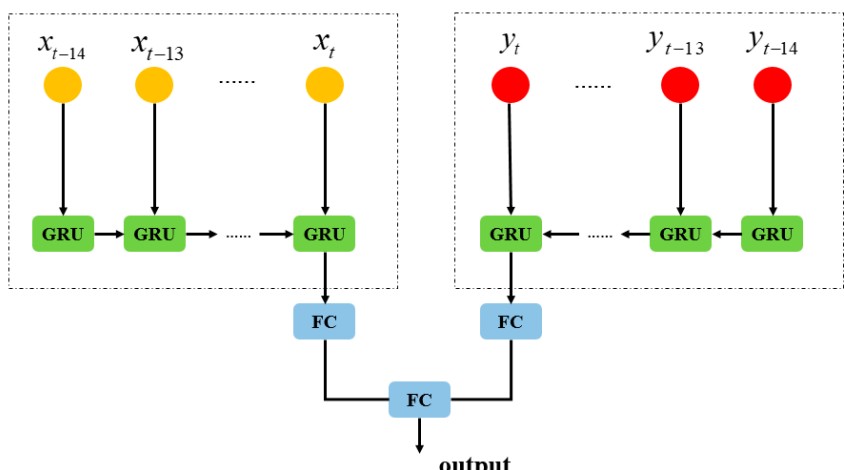

**Figure 2.** Proposed stock price prediction architecture.

In this model, both the target stock prediction module and the auxiliary module are trained using the GRU model. Each module produces an output through a fully connected layer. These two outputs are subsequently inputted into another fully connected layer to obtain the final output, representing the predicted price of the target stock. The algorithmic structure of the model is shown in Algorithm 1: The algorithm process for the improved GRU-based stock price prediction architecture.

| **Algorithm 1:** Improved GRU-based stock price prediction architecture |
|---|

**Input:** The historical data for the target stock $\{X_t\}_{t=1}^{N_i}$ and the historical data for 10 related stocks in the same industry $\{(Y1_t, Y2_t, Y3_t, Y4_t, Y5_t, Y6_t, Y7_t, Y8_t, Y9_t, Y10_t)\}_{t=1}^{N_i}$
**Output:** Predicted price of the target stock for the next day $x_{t+1}$
1    Initialize GRU model parameters;
2    **for** e ∈ (1,E) **do**
3      **for** t ∈ (1,$N_i$) **do**
4        $Y_t' = (y1_{t-14}, y1_{t-13}, \ldots, y1_t) =$
5        Fixed_selection $\{(Y1_t, Y2_t, Y3_t, Y4_t, Y5_t, Y6_t, Y7_t, Y8_t, Y9_t, Y10_t)\}_{t-14}^{t}$ or
6        Random_ selection $\{(Y1_t, Y2_t, Y3_t, Y4_t, Y5_t, Y6_t, Y7_t, Y8_t, Y9_t, Y10_t)\}_{t-14}^{t}$
7        $X_t' = [x_{t-14}, x_{t-13}, \ldots, x_t]$
8        gru_1 = GRU($X_t'$)
9        gru_2 = GRU($Y_t'$)
10        $x_{t+1}$ = concatenated (gru_1, gru_2)
11      **end for**
12    **end for**
13    **return** $x_{t+1}$;

### 3.3. Datasets

Based on the formula for sample size selection, when N ≤ 3, one-by-one sampling is used; when 3 < N ≤ 300, random sampling is conducted with a sample size of $\sqrt{N} + 1$; when the total sample size N > 300, random sampling is conducted with a sample size of $\frac{\sqrt{N}}{2} + 1$.

In this experiment, stock data were chosen based on the industry categorization. An L2 industry typically contains only around 100 samples. Hence, this study employed a sampling quantity equation to determine the selection of sample stock data. The sample size derived from the formula for the selected industry was less than or just over 10. To ensure experimental consistency, 10 stocks from the same industry as the target stock to be predicted were selected in each experiment. Using these 10 stocks, results similar to those generated by the entire industry can be achieved. In this experiment, the following five parameters of the stock were considered: opening price (open), highest price (high), lowest price (low), closing price (close), and trading volume (vol). These 10 stocks were chosen to represent the trend for the entire industry, as their features had a relatively significant impact on the information for the target stock to be predicted in this study.

The dataset chosen for the research encompassed industries closely related to our daily lives, including the liquor, pharmaceutics, banking, and film and television industries. We utilized the open-source database tushare to acquire historical data for representative companies in the aforementioned industries, as well as 10 related companies within the same industries, during the period from 10 April 2018 to 23 December 2022. This dataset spanned a total of 1146 trading days, encompassing the historical data for the five parameters mentioned above. Subsequently, we utilized these data to forecast the stocks of the respective representative companies. Details of the target stocks predicted and the selected related stocks are shown in the Table 1.

**Table 1.** Target stocks and related stocks.

| Industry | Target Stocks | Related Stocks |
|---|---|---|
| Distilled liquor | Gujing Gongjiu | Maotai Guizhou, Wuliangye, Yanghe Distillery, Luzhou Laojiao, Fenjiu, Shunxin Agriculture, Jinshiyuan, Kouzi Jiu, Shui-jingfang, Yingjia Gongjiu, Jiuguijiu |
| Pharmaceutics | Laobaixing | Shanghai Pharmaceutical, Huadong Medicine, Jiuzhou Tong, Da Can Lin, China National Pharmaceutical Group Corp., China National Prescription Drug Co., Ltd., China Medical System Holdings Limited, Haiwang Biology Co., Ltd., Yi Xin Tang, Taiyangneng |

**Table 1.** *Cont.*

| Industry | Target Stocks | Related Stocks |
|---|---|---|
| Banking | Bank of Communications | Industrial and Commercial Bank of China (ICBC), China Construction Bank (CCB), Agricultural Bank of China (ABC), Bank of China (BOC), China Merchants Bank (CMB), Industrial Bank Co Ltd (IB), Shanghai Pudong Development Bank (SPDB), Ping An Bank, China CITIC Bank, China Minsheng Banking Corp Ltd. |
| Cinema | Dongyanghengdian Film and Television City | Enlight Media, China Film Group Corporation, Huace Film and TV, Alpha Group Co., Ltd., Huayi Brothers Media Corp, Beijing Culture Co., Ltd., Central Motion Picture Corporation, H&R Century Pictures Co., Ltd., Shanghai Film Group Corporation, Bona Film Group Limited |

*3.4. Normalization*

The notable differences observed among the five parameters of each type of stock could have impacted the optimization of the trained model weights in the later stages. To eliminate such impacts, the present study employed a normalization formula (Formula (5)) to standardize the data within the range of [0, 1].

$$X_i^n = \frac{X_i - X_{\min}}{X_{\max} - X_{\min}} \tag{5}$$

*3.5. Construction of the Auxiliary Module Dataset*

The number of data points to be fed into the proposed model was only 1146. Such a small amount of data would result in overfitting when used to train a deep learning model. To overcome the issue, data augmentation was applied with the deep neural networks presented in the survey [33]. This paper proposes a data augmentation method to effectively reconstruct datasets [9]. The core focus of this model lies in the construction of the auxiliary module dataset. Ten stocks were chosen by using the aforementioned methods and subsequently normalized. For this experiment, two distinct approaches were employed to handle the historical data for these 10 stocks.

In this study, three models were trained, details of which are presented in Table 2. The first was the GRU model. This approach utilized a single prediction module and the model was trained without an auxiliary module. Secondly, five fixed stocks were randomly selected from the ten chosen stocks, and the average of their respective parameters was used to create the dataset required for the auxiliary module. The approach was named StockAugNet-f. Another approach was to randomly select five stocks at a time from the ten chosen stocks for each trading day. The average of their respective parameters was used to create the dataset required for the auxiliary module. This was the network of interest, which was termed StockAugNet-c. This process is illustrated in Figure 3.

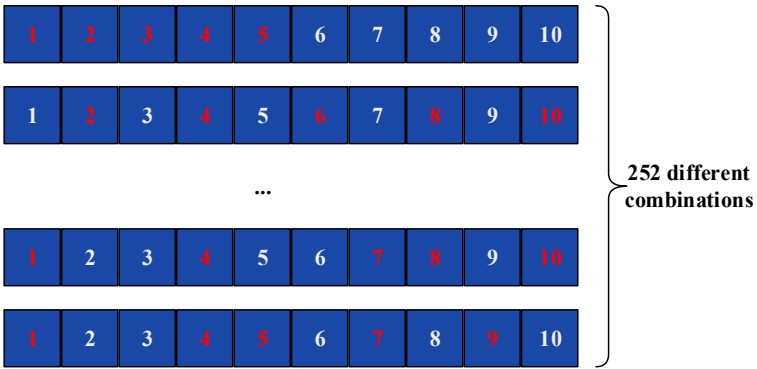

**Figure 3.** Data selection method diagram.

**Table 2.** Proposed model input data.

| Model | Auxiliary Module | Prediction Module |
|---|---|---|
| GRU | Not present | |
| StockAugNet-f | The historical data for fixed five stocks | The historical data for the target stock |
| StockAugNet-c | Combination of the 10 stocks taken 5 at a time | |

As this study focused on predicting target stocks, the historical data for the target stock were the main input. In this experiment, a rolling time window of 15 was set, which meant that the historical data for the first 15 lagged days were used as input, and the data for the 16th day were used as output. Then, following this pattern, the data for the 2nd to 16th day were used as input, and the data for the 17th day were used as output. This process is illustrated in the figure. To validate the effectiveness of the model, the window size can also be set to 5, 10, or 20 days during the experiment. In this specific experiment, a window size of 15 days was used to assess the model's performance.

*3.6. Evaluation Parameter*

To evaluate the model prediction results, this article selected three evaluation metrics: the root mean square error (RMSE), mean absolute error (MAE), and mean absolute percentage error (MAPE). The calculation methods for each evaluation metric are as follows.

(1) RMSE

The RMSE is commonly utilized to assess the extent of deviation between predicted outcomes and actual data. A smaller RMSE value indicates a higher accuracy for the prediction model. The RMSE is mathematically defined as follows:

$$\text{RMSE} = \sqrt{\frac{1}{N}\sum_{i=1}^{N}(y_i - y'_i)^2} \tag{6}$$

(2) MAE

The MAE refers to the average absolute deviation between the arithmetic mean and individual observed values. A smaller MAE value indicates higher prediction accuracy. The MAE is mathematically defined as follows:

$$\text{MAE} = \frac{1}{N}\sum_{i=1}^{N}\left(|y_i - y'_i|\right) \tag{7}$$

(3) MAPE

The MAPE is utilized to quantify the average deviation between the predicted value and the actual value. A lower MAPE value implies a higher level of prediction accuracy. The MAPE is mathematically defined as follows:

$$\text{MAPE} = \sum_{i=1}^{N}\left|\frac{y_i - y'_i}{y_i}\right| \times \frac{100}{N} \tag{8}$$

## 4. Results

This section presents experimental findings obtained by employing the model proposed in Section 3.2 to analyze stock data across multiple industries. We selected stock data from various industries that are pertinent to individuals' daily lives. These industries included the distilled liquor, pharmaceutics, banking, and cinema industries. The input datasets for the two modules consisted of historical data for representative company stocks from various industries and a dataset constructed using historical data for related company stocks within the same industry. The selected dataset included the opening price, closing price, highest price, lowest price, and trading volume for 1146 trading days from 10 April 2018

to 23 December 2022 for each company. The output was the predicted price of the target stock for the following day. We conducted experiments using three different methods: the classical GRU model, StockAugNet-f, and StockAugNet-c. A comparison of the results of these methods was undertaken.

### 4.1. The Experimental Results for the GRU

The GRU was the basic network trained without the auxiliary module. The RMSE of the network was 0.263, 0306, 0.242, and 0.205 for each of the four industries. In the loss function graph for the GRU model, it can be observed that the training set achieved high accuracy across the four industries, while the validation set exhibited larger errors, as shown in Figure 4.

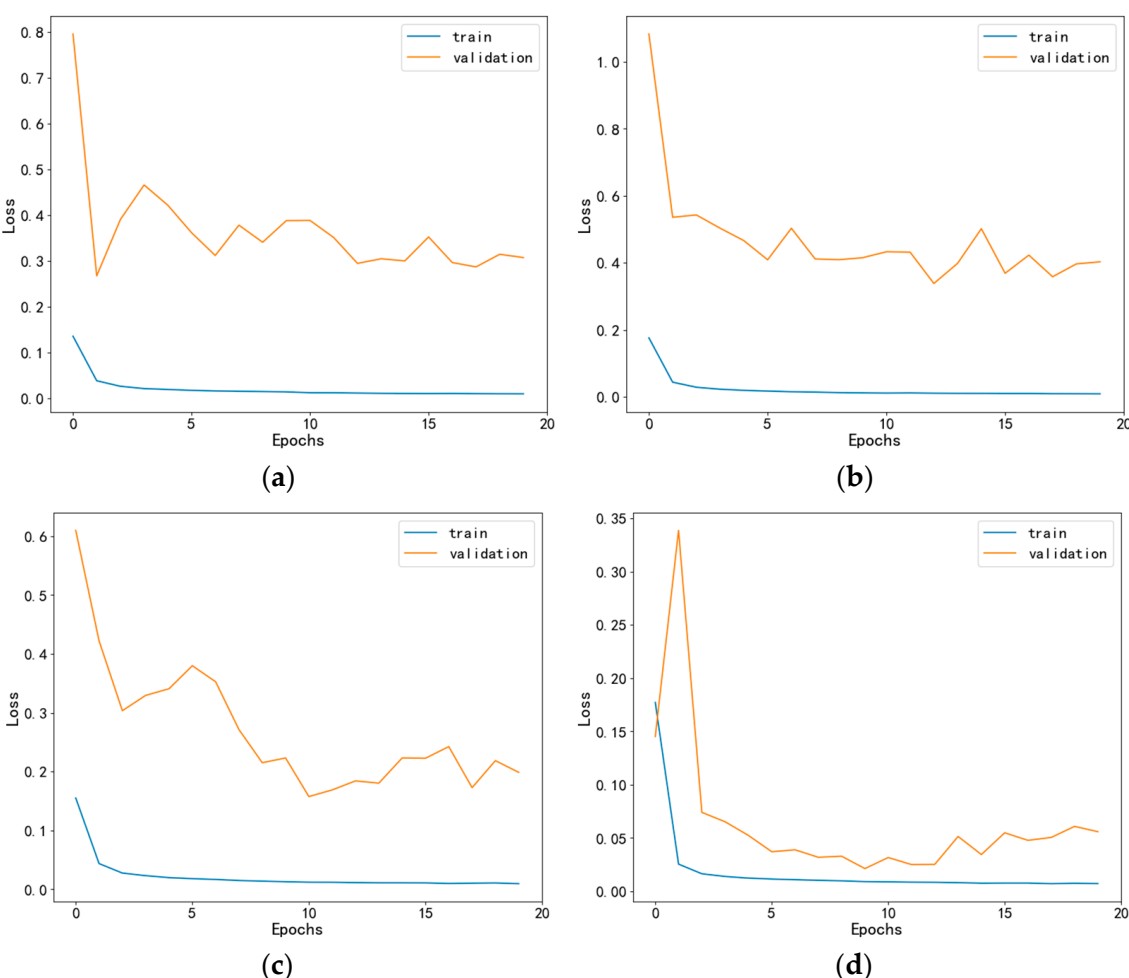

**Figure 4.** (**a**) Loss function error plot for GRU for distilled liquor industry; (**b**) loss function error plot for GRU for pharmaceutics industry; (**c**) loss function error plot for GRU for banking industry; (**d**) loss function error plot for GRU for cinema industry.

### 4.2. The Experimental Results for StockAugNet-f

In this section, we analyze the prediction results obtained using StockAugNet-f; i.e., with 5 fixed stocks in the auxiliary module. Compared to the GRU model, the StockAugNet-f model had five times the number of input parameters. As shown in Figure 5, in the loss graphs across different industries, it can be observed that, compared to the single GRU model, the StockAugNet-f model achieved significant improvements in accuracy for both the training and test sets. Additionally, in the price prediction trend graphs, depicted in Figures 6–9, we can observe that the results predicted by this model with the test set exhibited a good fit with the truth values.

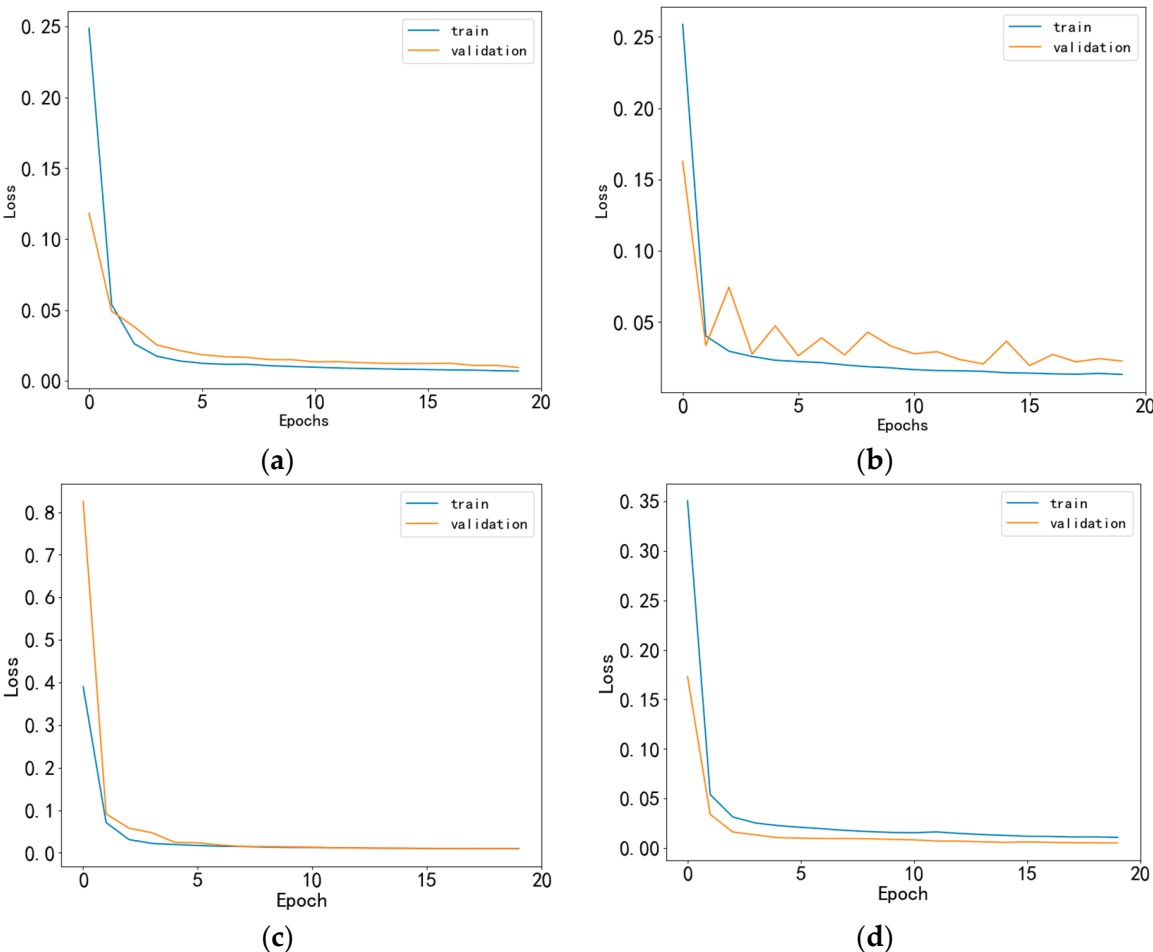

**Figure 5.** (**a**) Loss function error plot for StockAugNet-f for distilled liquor industry; (**b**) loss function error plot for StockAugNet-f for pharmaceutics industry; (**c**) loss function error plot for StockAugNet-f for banking industry; (**d**) loss function error plot for cinema industry.

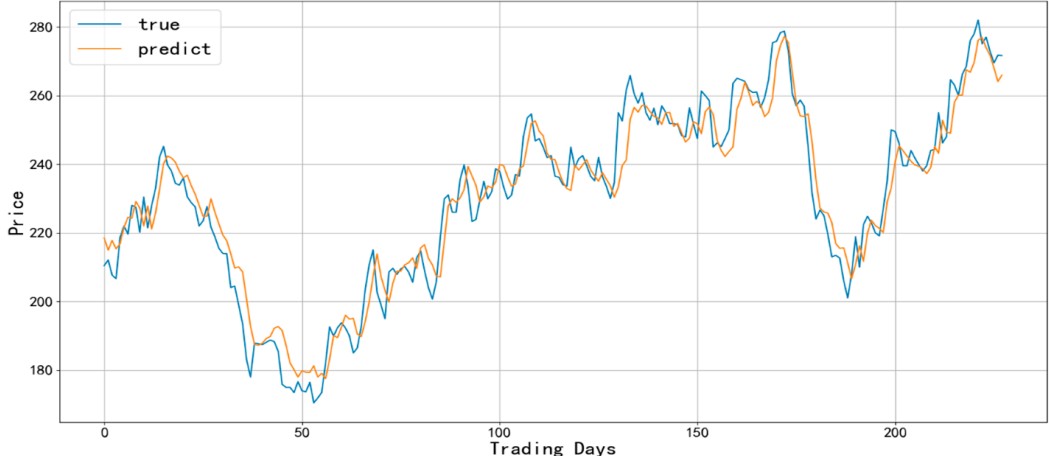

**Figure 6.** Comparison between the predicted and actual values for distilled liquor industry (StockAugNet-f).

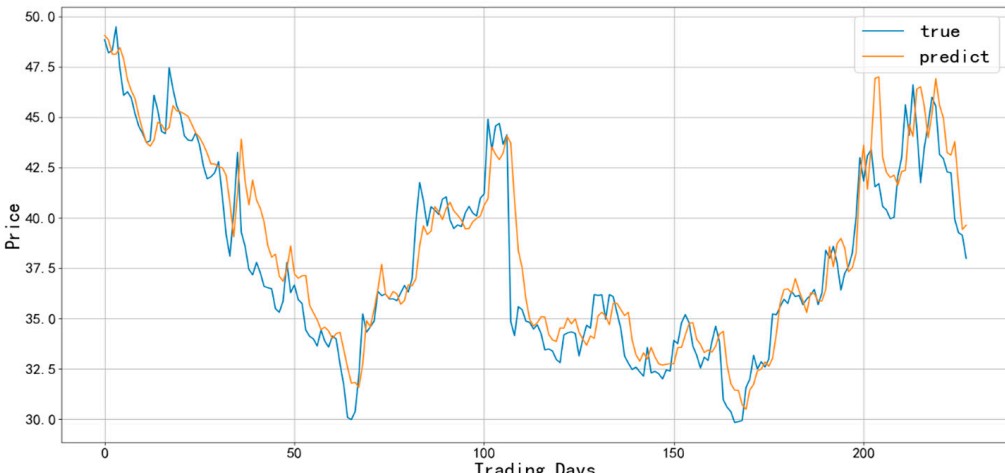

**Figure 7.** Comparison between the predicted and actual values for the pharmaceutics industry (StockAugNet-f).

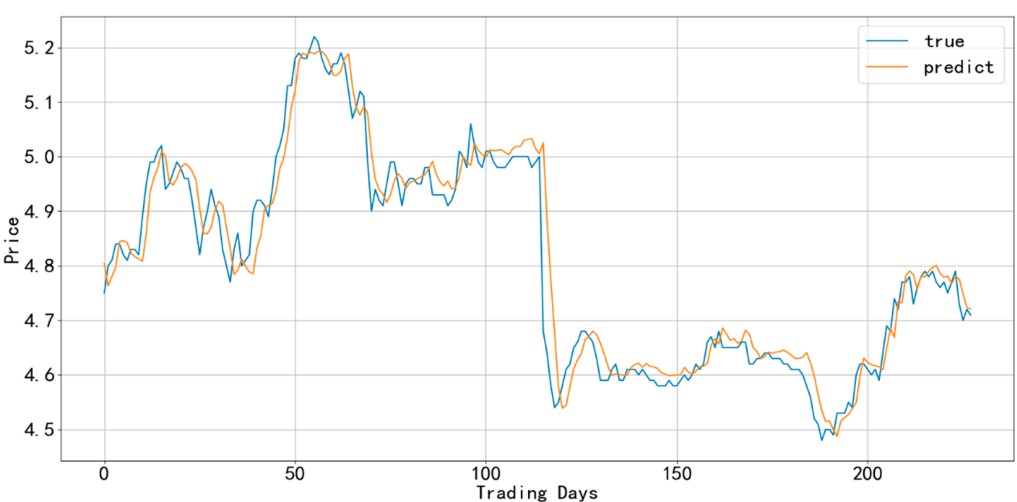

**Figure 8.** Comparison between the predicted and actual values for the banking industry (StockAugNet-f).

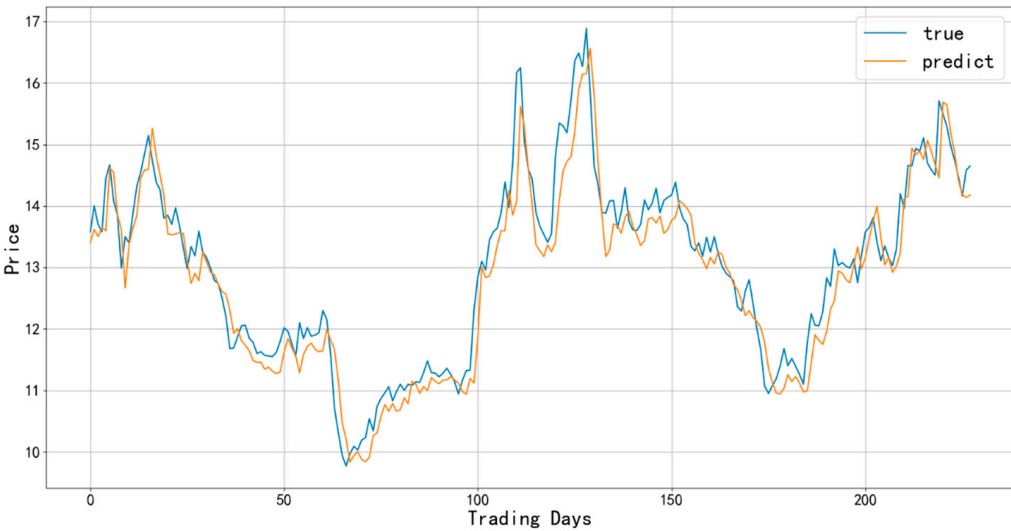

**Figure 9.** Comparison between the predicted and actual values for the cinema industry (StockAugNet-f).

### 4.3. The Experimental Results for StockAugNet-c

In this section, we analyze the prediction results obtained using StockAugNet-c; i.e., taking 5 stocks at a time from the 10 chosen stocks for each trading day. The loss function graph for the StockAugNet-c model with the four industries is shown in Figure 10. From the graph, it can be observed that the accuracy of the test set was significantly improved compared to the GRU model. Additionally, there was a slight improvement in accuracy compared to the StockAugNet-f model.

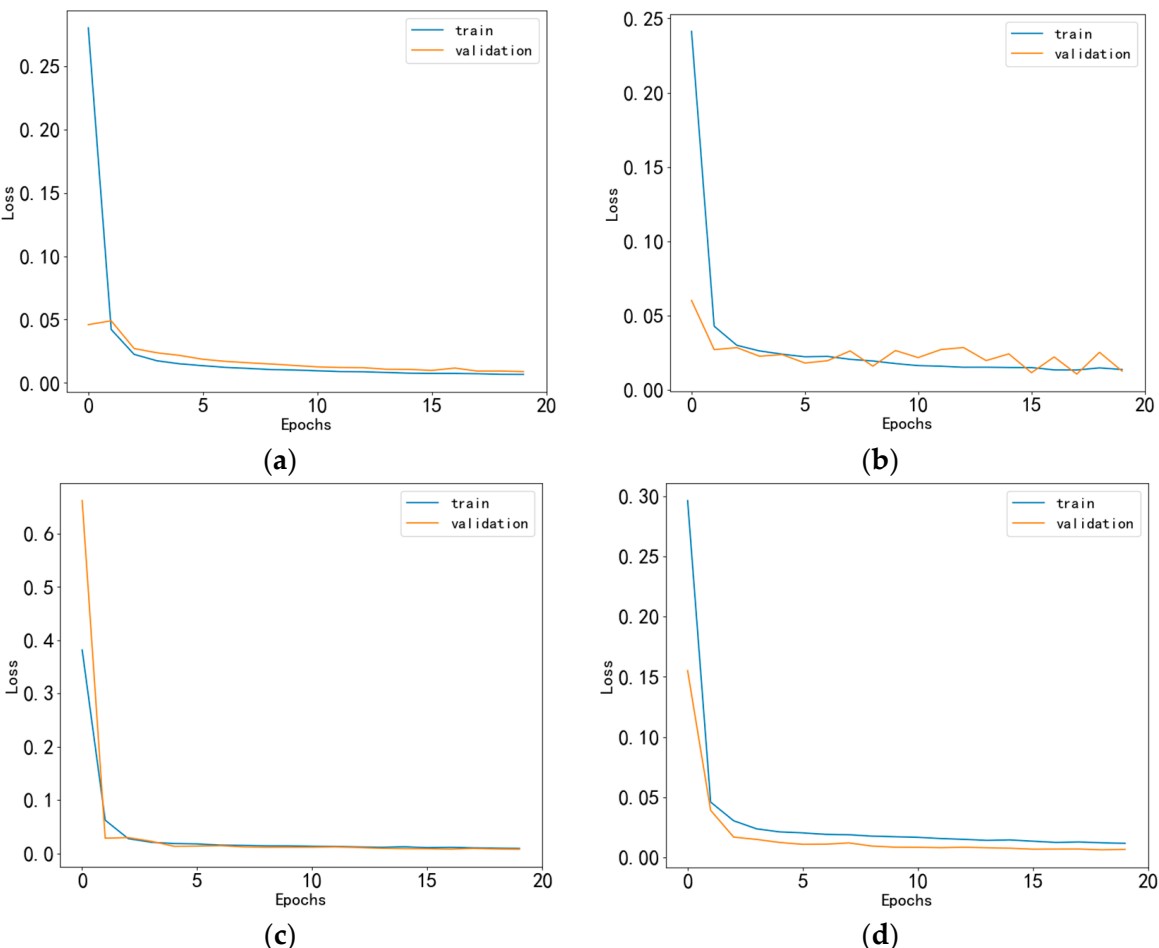

**Figure 10.** (**a**) Loss function error plot for StockAugNet-c for distilled liquor industry; (**b**) loss function error plot for StockAugNet-c for pharmaceutics industry; (**c**) loss function error plot for StockAugNet-c for banking industry; (**d**) loss function error plot for StockAugNet-c for cinema industry.

For the four industries, the StockAugNet-c model achieved RMSE values of 0.101, 0.126, 0.084, and 0.075, which were substantial improvements compared to the GRU model's RMSE values of 0.263, 0.306, 0.242, and 0.205. Although the model's output primarily depends on the input from the prediction module, the auxiliary module reduces the chance of overfitting and thus improves the prediction accuracy. Figures 11–14 display a comparison between the predicted values obtained with the StockAugNet-c model with different industry input data and the truth values.

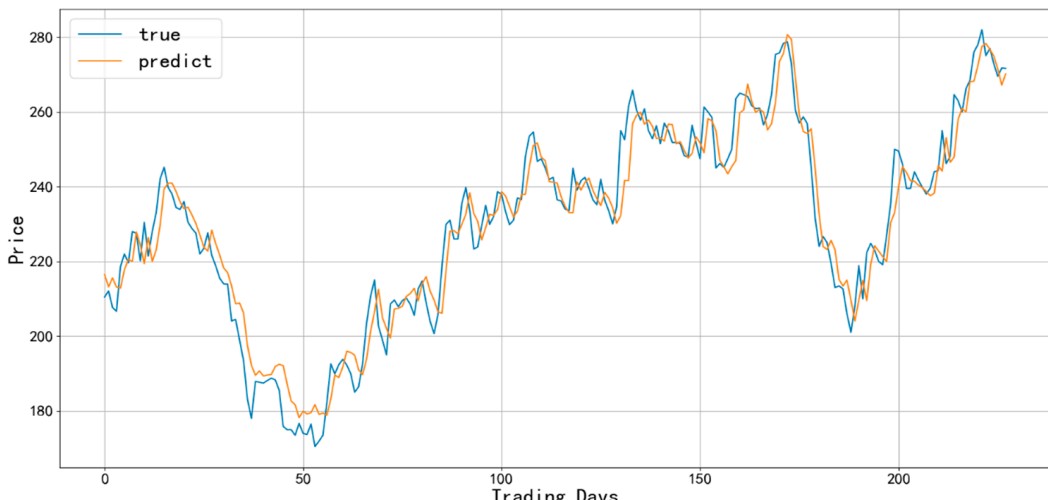

**Figure 11.** Comparison between the predicted and actual values for distilled liquor industry (StockAugNet-c).

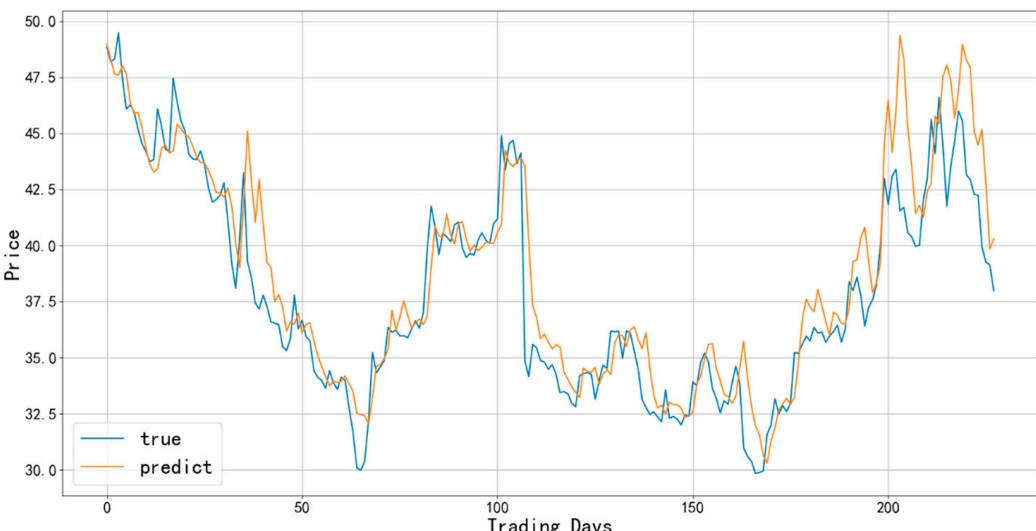

**Figure 12.** Comparison between the predicted and actual values for the pharmaceutics industry (StockAugNet-c).

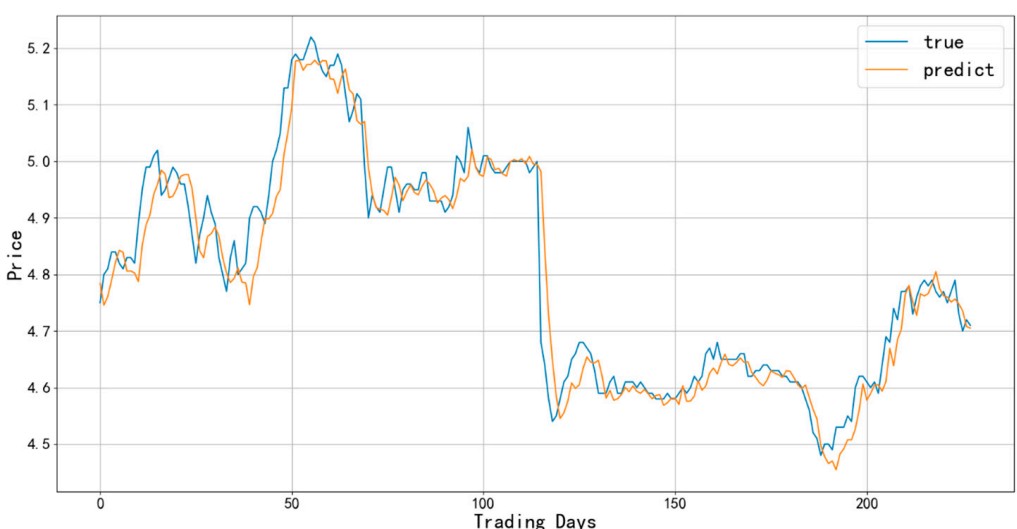

**Figure 13.** Comparison between the predicted and actual values for the banking industry (StockAugNet-c).

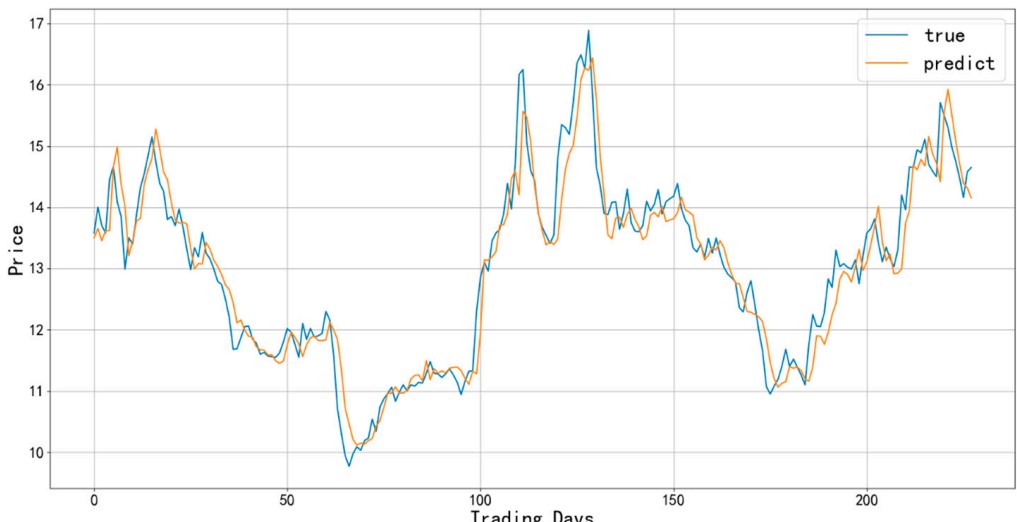

**Figure 14.** Comparison between the predicted and actual values for the cinema industry (StockAugNet-c).

*4.4. Comparison of Experimental Results*

In this section, we analyze the performance of the proposed approach using the three evaluation parameters; namely, the RMSE, MAE, and MAPE, as presented in Section 3.6.

We compared our proposed model with four baseline models; namely, a GRU, LSTM, an RNN, and a DNN. All six models were provided with input consisting of stock index data from the same four industries over the same time period. Three evaluation metrics—namely, the RMSE, MAE, and MAPE—were used to validate the models' output against the actual prices. The comparison results are depicted in Tables 3–5 and in Figures 15–17.

**Table 3.** The RMSEs for the four industries with six different experimental methods.

|  | Distilled Liquor | Pharmaceutics | Banking | Cinema |
|---|---|---|---|---|
| StockAugNet-f | **0.089** | 0.144 | 0.095 | 0.082 |
| StockAugNet-c | 0.101 | **0.126** | **0.084** | **0.075** |
| GRU | 0.263 | 0.306 | 0.242 | 0.205 |
| LSTM | 0.273 | 0.312 | 0.261 | 0.268 |
| RNN | 0.298 | 0.353 | 0.288 | 0.322 |
| DNN | 0.386 | 0.440 | 0.368 | 0.355 |

**Table 4.** The MAEs for the four industries with six different experimental methods.

|  | Distilled Liquor | Pharmaceutics | Banking | Cinema |
|---|---|---|---|---|
| StockAugNet-f | 26.23 | 12.12 | 32.57 | 7.92 |
| StockAugNet-c | **25.89** | **9.31** | **22.94** | **7.03** |
| GRU | 38.21 | 12.60 | 38.08 | 9.64 |
| LSTM | 46.48 | 19.83 | 43.29 | 11.24 |
| RNN | 113.21 | 69.41 | 116.17 | 44.16 |
| DNN | 140.33 | 142.05 | 174.54 | 122.32 |

**Table 5.** The MAPEs for the four industries with six different experimental methods.

|  | Distilled Liquor | Pharmaceutics | Banking | Cinema |
|---|---|---|---|---|
| StockAugNet-f | 0.263 | 0.105 | 0.371 | 0.099 |
| StockAugNet-c | **0.262** | **0.081** | **0.259** | **0.092** |
| GRU | 0.324 | 0.126 | 0.403 | 0.107 |
| LSTM | 0.524 | 0.226 | 0.462 | 0.198 |
| RNN | 0.84 | 0.51 | 1.07 | 0.41 |
| DNN | 1.34 | 1.41 | 1.71 | 1.19 |

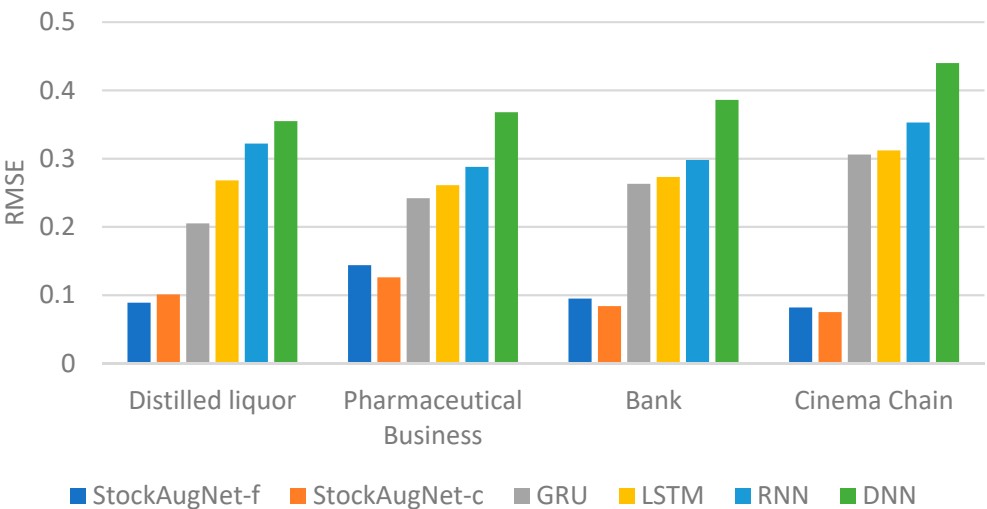

**Figure 15.** The RMSEs for the four industries with six different experimental methods.

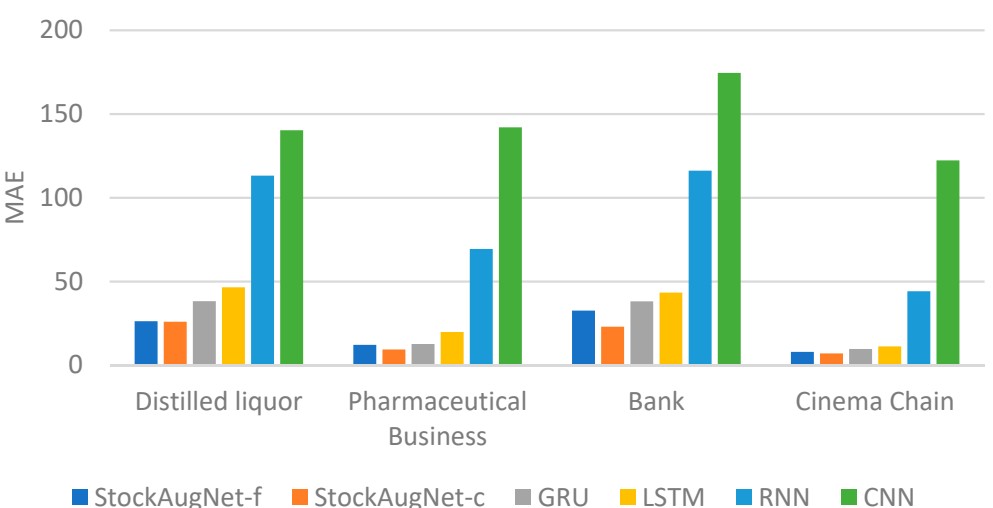

**Figure 16.** The MAEs for the four industries with six different experimental methods.

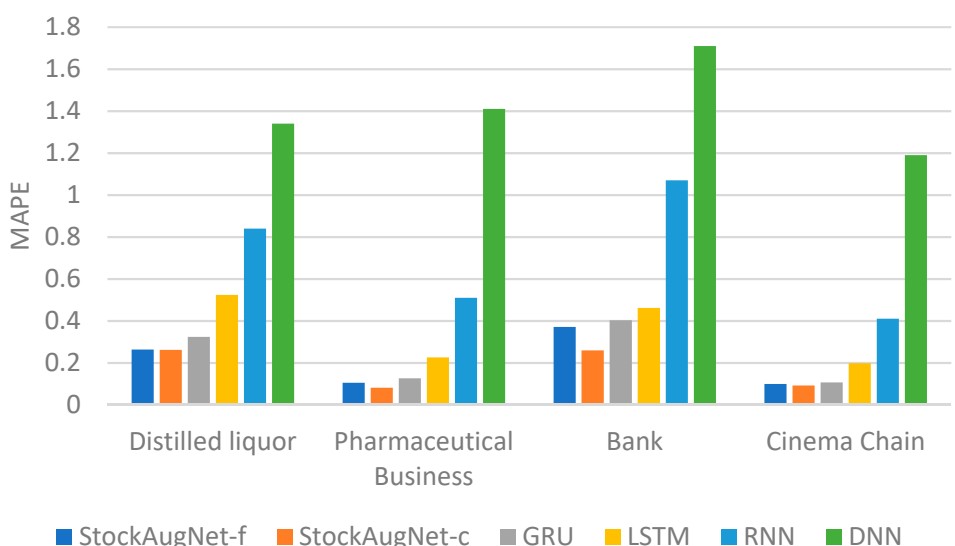

**Figure 17.** The MAPEs for the four industries with six different experimental methods.

As depicted in the figures and tables, this study introduced two methods that exhibited superior performance compared to the other four methods across all three metrics (the RMSE, MAE, and MAPE). Particularly in terms of the RMSE, both StockAugNet-f and StockAugNet-c achieved error reductions of more than twofold compared to the GRU model, which was the best-performing model among the remaining four models. Furthermore, in terms of the MAE and MAPE, StockAugNet-c demonstrated the best performance across all four industries. Specifically, for the banking industry, StockAugNet-c exhibited reductions in the MAE of 15.14, 20.35, 93.23, and 151.6 percentage points compared to the other four models, while the corresponding reductions in the MAPE were 0.144, 0.203, 0.811, and 1.451. Although StockAugNet-f performed slightly worse than StockAugNet-c, it still showed significant improvements compared to the other four models.

Based on the experimental findings, it is evident that the stock prediction model proposed in this paper, along with the data reconstitution method, achieved significant decreases in the loss function values across multiple industry-specific training and validation sets. Notably, even in the pharmaceutics sector, where slight fluctuations were observed, the performance of the model remained commendable. These results underscore the effectiveness of the experimental methodology in mitigating overfitting, thereby demonstrating its exceptional generalization capacity and robustness.

## 5. Discussion

Stock price prediction has always been a passionate area of research, and the research methods have constantly evolved along with technological advancements. The initial approaches in this field were based on econometric methods, which were later replaced by machine learning methods and, more recently, deep learning techniques. In the realm of stock price prediction research, it is common to encounter small datasets. Classic methods, such as DNNs, RNNs, LSTM, and GRUs, as mentioned above, often suffer from overfitting issues when applied to small datasets. To address this problem, this article presented a GRU-based stock prediction model using a reconstructed dataset. Traditional approaches to tackle the low data volume issue in stock price prediction involve data augmentation. However, these augmentation methods are solely based on the historical data for the target stock without taking into consideration the influence of industry-specific characteristics on stock prices. In contrast, this study innovatively utilized highly correlated stocks within the same industry as an auxiliary module to adjust the training results of the prediction module. This approach offers two advantages: firstly, it increases the amount of data, expanding the scale of training samples and enhancing the training comprehensiveness, resulting in improved generalization capabilities; secondly, it enriches the features extracted by the model, reducing the risk of overfitting and improving the accuracy of predictions.

Additionally, the method for selecting stocks from different industries also has varying effects on the improvement of prediction accuracy. Random selection methods enable a model to capture a broader range of data features, thereby enhancing its generalizability. On the other hand, fixed selection methods yield more stable prediction results, but the limitation of the data selection may restrict the capability to express overall industry features. Our experimental results also demonstrated that our proposed model improved the accuracy of the test set without adversely affecting the accuracy of the training set compared to the previously mentioned deep learning methods, such as the DNN, the RNN, LSTM, and the GRU, all of which effectively improved the predictive performance of the model to varying degrees.

Therefore, the proposed model based on the reconstructed dataset offers a new perspective for stock prediction research. It holds positive implications not only for researchers and investors with regard to stock valuation and risk assessment but also for decision makers in the financial field who seek more accurate and reliable decision-making guidance. By leveraging richer data features and reducing the risk of overfitting, our model is better equipped to adapt to market variations and can provide more precise predictions. This has important implications for investors in formulating investment strategies and managing

risks, while also offering beneficial exploration for the academic community in the realm of stock prediction and financial data analysis.

In future research, we plan to explore various other deep learning networks or combinations of deep learning networks with recently developed data augmentation techniques, as used in image augmentation [34]. Additionally, the model will also be trained using many more input features, such as financial news data and sentiment analysis [35], as these can serve as supplementary sources of information.

**Author Contributions:** Conceptualization, C.C. and W.X.; methodology, C.C.; software, C.C.; validation, C.C., L.X. and W.X.; formal analysis, C.C.; data curation, W.X.; writing—original draft preparation, C.C.; writing—review and editing, W.X., C.C. and L.X.; visualization, W.X.; supervision, L.X.; project administration, W.X. All authors have read and agreed to the published version of the manuscript.

**Funding:** This research received no external funding.

**Institutional Review Board Statement:** Not applicable.

**Informed Consent Statement:** Not applicable.

**Data Availability Statement:** Not applicable.

**Conflicts of Interest:** The authors declare no conflict of interest.

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
