# Peer review of "Research on Improved GRU-Based Stock Price Prediction Method"

_applsci, doi:10.3390/app13158813_

Round 1

Reviewer 1 Report

Research on Improved GRU-based Stock Price Prediction Method is presented and the following are the key issues that need to be addressed

Since there are plenty of works associated with stock price prediction in the past, it is necessary to highlight the novelty involved in this work strongly.

The key novel contributions need to be illustrated separately in the introduction part.

Figure 1 is unnecessary

The proposed model should be presented in an algorithmic way

Why only two approaches are chosen for the comparison of results?

The other proven techniques also should have been considered for the comparison of results.

As the availability of plenty of proven models already, the results analysis section does not look convincing and the need for the proposed model needs to be strongly justified.

Author Response

Dear Reviewer,

Thank you so much for reviewing our manuscript and giving us the chance to revise our manuscript. Based on the valuable comments and suggestions, we have revised the manuscript seriously. The important revisions have been provided in the revised manuscript. The necessary explanations are also provided in this point-by-point response letter. We hope our explanations and revisions could address all the concerns. Thanks a lot again.

Best regards,

Chi Chen

July 2023

Reviewer 2 Report

Dear authors,

First, thank you for the opportunity to review this paper, which covers a topic of significant interest and novelty. Despite that, I would suggest you improve the project based on the following comments, some of them intending to increase the paper's readability:

1. some paragraphs are too long and difficult to read without losing our focus.

2. It is not clear in the introduction the novelty that the paper raises in comparison with previous literature, particularly as regards the methods used.

3. Results are not significantly described. The figures and charts appear with no relevant analysis that can be helpful to users be guided. As a minor suggestion, I think that the figures shown in some graphs are redundant.

4. This is also a relevant matter within the discussion, which is significantly poor. Readers are unaware of the benefits of the findings from this paper, based on previous literature using the same or different methods. How they are comparable or not? What are its novelty, significance and contributions? In what sense this paper can contribute to research, professional, entities, etc.?

To summarise, I consider that the paper must be significantly improved to convince readers how valuable it is.

I hope those suggestions may be useful.

Please see the comments above on this aspect also.

Author Response

Dear Reviewer,

Thank you so much for reviewing our manuscript and giving us the chance to revise our manuscript. Based on the valuable comments and suggestions, we have revised the manuscript seriously. The important revisions have been provided in the revised manuscript. The necessary explanations are also provided in this point-by-point response letter. We hope our explanations and revisions could address all the concerns. Thanks a lot again.

Round 2

Reviewer 1 Report

Authors have addressed all the major suggestions given and this work can be accepted now. 

Author Response

Dear Reviewer,

Thank you for your previous comments on my manuscript and for the acceptance of my manuscript this time. It is precisely because of your suggestions that the quality of my manuscript has been greatly improved. Thanks a lot again.

Best regards,

Chi Chen

July 2023

Reviewer 2 Report

Dear authors,

Thanks for the significant efforts made to the revised version of the paper, which highly improved its overall quality. The only point that I think could be still improved is related to the discussion, where I would expect to see a better link with previous literature shown in sections 1 and 2. Nonetheless, I will assume that authors can do that in the final version to be sent. Therefore, it is Ok for me to accept the paper at this stage with this minor suggestion for improvement.

No relevant issue to highlight on this topic.

Author Response

Dear Reviewer,

Thank you so much for reviewing our manuscript and giving your valuable comments. We made careful revisions in the final version of the manuscript based on your valuable suggestions. The necessary explanations are also provided in the response letter. We hope our explanations and revisions could address the concern. Thanks a lot again.

Best regards,

Chi Chen

July 2023
